# The mosquito vectors that sustained malaria transmission during the Magude project despite the combined deployment of indoor residual spraying, insecticide-treated nets and mass-drug administration

Lucía Fernández Montoya[1,2]*, Helena Martí-Soler[1], Mara Máquina[2], Kiba Comiche[2], Inocencia Cuamba[2], Celso Alafo[2], Lizette L. Koekemoer[3], Ellie Sherrard-Smith[4], Quique Bassat[1,2,5,6,7], Beatriz Galatas[1,2¤], Pedro Aide[2,8], Nelson Cuamba[9,10], Dulcisaria Jotamo[9], Francisco Saúte[2], Krijn P. Paaijmans[1,2,11,12,13]

1 ISGlobal, Barcelona, Spain, 2 Centro de Investigação em Saúde de Manhiça (CISM), Fundação Manhiça, Mozambique, 3 Faculty of Health Sciences, WITS Research Institute for Malaria, University of the Witswatersrand and the Natitonal Institute for Communicable Diseases, Johannesburg, South Africa, 4 MRC Centre for Global Infectious Disease Analysis, Imperial College London, London, United Kingdom, 5 ICREA, Barcelona, Spain, 6 Pediatrics Department, Hospital Sant Joan de Déu, Universitat de Barcelona, Esplugues, Barcelona, Spain, 7 Consorcio de Investigación Biomédica en Red de Epidemiología y Salud Pública (CIBERESP), Madrid, Spain, 8 Instituto Nacional da Saúde, Ministério da Saúde, Maputo, Mozambique, 9 Programa Nacional de Controlo da Malária, Ministério da Saúde, Maputo, Mozambique, 10 PMI VectorLink Project, Abt Associates Inc., Maputo, Mozambique, 11 Center for Evolution and Medicine, School of Life Sciences, Arizona State University, Tempe, AZ, United States of America, 12 The Biodesign Center for Immunotherapy, Vaccines and Virotherapy, Arizona State University, Tempe, AZ, United States of America, 13 Simon A. Levin Mathematical, Computational and Modeling Sciences Center, Arizona State University, Tempe, AZ, United States of America

¤ Current address: Global Malaria Program, World Health Organization, Geneva, Switzerland
* lucia.fmontoya@gmail.com

## Abstract

The "Magude project" aimed but failed to interrupt local malaria transmission in Magude district, southern Mozambique, by using a comprehensive package of interventions, including indoor residual spraying (IRS), pyrethroid-only long-lasting insecticide treated nets (LLINs) and mass-drug administration (MDA). Here we present detailed information on the vector species that sustained malaria transmission, their association with malaria incidence and behaviors, and their amenability to the implemented control interventions. Mosquitoes were collected monthly between May 2015 and October 2017 in six sentinel sites in Magude district, using CDC light traps both indoors and outdoors. *Anopheles arabiensis* was the main vector during the project, while *An. funestus s.s.*, *An. merus*, *An. parensis* and *An. squamosus* likely played a secondary role. The latter two species have never previously been found positive for *Plasmodium falciparum* in southern Mozambique. The intervention package successfully reduced vector sporozoite rates in all species throughout the project. IRS was effective in controlling *An. funestus s.s.* and *An. parensis*, which virtually disappeared after its first implementation, but less effective at controlling *An. arabiensis*. Despite suboptimal use, LLINs likely provided significant protection against *An. arabiensis* and *An. merus* that

**Data Availability Statement:** All relevant data are within the paper and its Supporting Information files.

**Funding:** This study was supported by the Bill and Melinda Gates Foundation and Obra Social "la Caixa" Partnership for the Elimination of Malaria in Southern Mozambique (INV-008483). LLK is supported by a DST/NRF South African Research Chairs Initiative Grant (UID 64763). ESS is funded by a UKRI Future Leaders Fellowship from the Medical Research Council (MR/T041986/1) and acknowledges funding from the MRC Centre for Global Infectious Disease Analysis (reference MR/R015600/1), jointly funded by the UK Medical Research Council (MRC) and the UK Foreign, Commonwealth & Development Office (FCDO), under the MRC/FCDO Concordat agreement and is also part of the EDCTP2 programme supported by the European Union; and acknowledges funding by Community Jameel. Abt. Associates Inc. provided support in the form of salaries for author NC, but did not have any additional role in the study design, data collection and analysis, decision to publish, or preparation of the manuscript. This does not alter our adherence to PLOS ONE policies on sharing data and materials. The specific role of this author is articulated in the 'author contributions' section.

**Competing interests:** The authors have declared that no competing interests exist.

sought their host largely indoors when people where in bed. Adding IRS on top of LLINs and MDA likely added value to the control of malaria vectors during the Magude project. Future malaria elimination attempts in the area could benefit from i) increasing the use of LLINs, ii) using longer-lasting IRS products to counteract the increase in vector densities observed towards the end of the high transmission season, and iii) a higher coverage with MDA to reduce the likelihood of human infection. However, additional interventions targeting vectors that survive IRS and LLINs by biting outdoors or indoors before people go to bed, will be likely needed to achieve local malaria elimination.

## Introduction

Despite the remarkable reductions in the malaria burden in sub-Saharan Africa over the last two decades [1,2], no country in this region has managed to eliminate malaria. The lowest malaria burden of all sub-Saharan Africa is observed in its southern part, namely in Namibia, Botswana, South Africa and eSwatini. In 2016, the World Health Organization (WHO) determined that eSwatini and South Africa had the potential to achieve zero indigenous cases by 2020. Albeit several regional malaria elimination efforts over the last few years [3,4], neither country was able to reach this target [5]. The importation of malaria cases from neighboring Mozambique, a country with considerably higher malaria transmission levels, has been highlighted as one of the causes [6]. In South Africa and eSwatini, malaria transmission is primarily concentrated in areas bordering Mozambique [5] and is driven by cases among migrant populations. Reducing or eliminating malaria in Mozambique, especially in its southern provinces, is therefore crucial to achieve malaria elimination in Southern African and eSwatini. The southern part of Mozambique has been targeted by initiatives to stop malaria transmission since the 1960's, but, unfortunately, none have led to local malaria elimination [6–9].

The first attempt to eliminate malaria in southern Mozambique took place between 1960 and 1969 in the context of the Global Malaria Eradication Program (GMEP) [7]. The second initiative, the Lubombo Spatial Development Initiative (LSDI), was implemented between 1999 to 2011 [8–10]. Both were based on indoor residual spraying (IRS), which aims to kill mosquitoes resting on walls and ceilings with insecticides, although the second elimination attempt combined IRS with targeting the parasite reservoir using artemisinin combination therapies (ACTs). More recently, in 2015, the Mozambique, South Africa and eSwatini (MOSASWA) regional initiative [6] and the Mozambican Alliance Towards the Elimination of Malaria (MALTEM) [9] were established. MOSASWA aimed to strengthen regional collaboration and efforts to accelerate progress towards achieving malaria elimination in the region. MALTEM aimed, among other objectives, to create the necessary knowledge to inform an operational elimination plan and roadmap for malaria elimination in Mozambique [9], which was piloted during the Magude project.

The Magude project was designed to evaluate the feasibility of eliminating malaria in southern Mozambique with a package of interventions available at the time, namely a combination of interventions targeting the vector (IRS and long-lasting insecticidal nets, or LLINs) and the parasite reservoir (mass drug administration, or MDA, and standard diagnosis and treatment) simultaneously. In addition, it implemented strong community engagement campaigns to maximize the acceptance and coverage of all interventions. The project was expected to reduce vector densities with a combination of the killing effects from IRS and LLINs, and then reduce transmission by surviving infectious vectors through the prophylactic effect of the MDA drugs

alongside the prevention of vector-human contact by LLINs, thereby closing the gap towards elimination [9].

While all of the initiatives listed above were successful at reducing the local burden of malaria [6–8], none of them achieved malaria elimination. Learning from past experiences is critical to guide future malaria elimination efforts in Mozambique and hence, to achieve elimination in the region. All these initiatives relied heavily on vector control, as do current and future malaria control efforts [11]. Therefore, 1) identifying the vectors that sustained malaria transmission despite the implemented vector control interventions and 2) evaluating the vectors' amenability to the implemented vector control products, are crucial to understanding the shortcomings of the piloted approaches and, hence, to guide the design of future malaria elimination efforts in southern Africa.

The outcome of the two aims above are presented here. We first describe anopheline species composition, densities, host-seeking behavior (time and place) and *P. falciparum* sporozoite rates during the course of the project. We then evaluate relative vector importance by exploring the association between densities of different vector species and malaria incidence, accounting for the implemented interventions. We subsequently combine these findings with previously published data on i) the efficacy of the three core interventions, LLINs, IRS and MDA [12], ii) susceptibility of the vectors to the used insecticides, and iii) the overlap between human and vector behaviors, to examine the ability of the implemented interventions to control the local vector populations. Finally, we use our new understanding to provide vector control recommendation for future malaria elimination efforts in the area.

## Materials and methods

### The Magude project in the Magude district

Magude district is a rural district located in Maputo province (southern Mozambique) that borders with Bilene district to the north and with Mpumalanga Province in South Africa to the West (Fig 1). The district has an area of 6,961 km$^2$ and is divided into five administrative posts, Magude Sede, Motaze, Panjane, Mahele and Mapulanguene. It had 48,448 residents in 2015 [13]. The vegetation is dominated by open forests and savannahs, and three main rivers cross the district. Most of the population relies on subsistence agriculture, fishing or working as sugar cane cutters in the sugar plantations in Magude, Xinavane or the neighboring Manhiça district. Houses are traditional round-shaped or rectangular-shaped huts constructed with cane, cement, mud brick or reeds and covered by adobe or cement. A comprehensive description of the district demographic, socio-economic and health characteristics, is provided elsewhere [13]. Two distinct climatological seasons are observed in southern Mozambique, a rainy season extending from October to March and a dry season from April to September. The high malaria incidence season occurs from November to April [13].

The Magude project started in 2015 by establishing a health and demographic platform to obtain the necessary information to guide the implementation of the core interventions during the project [13]. The district was covered (i.e. one net for every two people in a household) with LLINs that were distributed by the National Malaria Control Program (NMCP) through a mass distribution campaign in May 2014. Between August 2015 and December 2017, the project delivered an annual single-round of IRS with DDT and pirimiphos-methyl in the first year and only pirimiphos-methyl in the second and third year, and two annual rounds of MDA that were one month apart. During each MDA round, the de-facto population of Magude (including visitors but excluding infants <6 months, women in the first trimester of pregnancy or severely ill individuals) received a full 3-day course of dihydroartemisinin–piperaquine (DHAp). Community mobilization campaigns were implemented to increase the

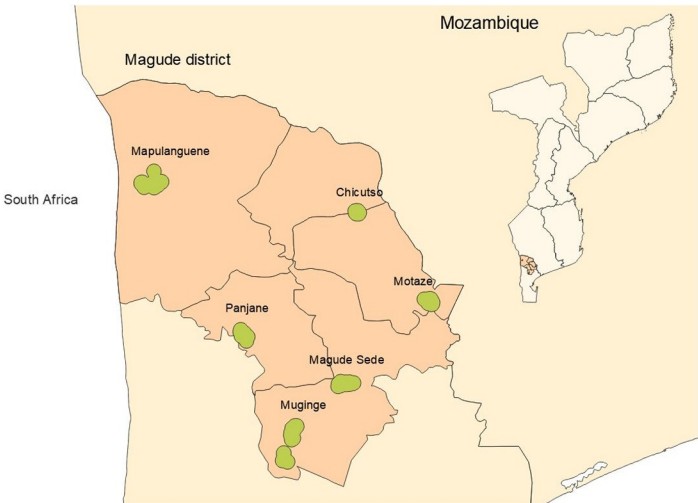

**Fig 1. Map of Magude district highlighting the administrative posts and sentinel sites for entomological surveillance.** Dots represent the houses where mosquitoes were collected. Map borders were obtained from the Humanitarian Data Exchange under license "Creative commons attribution for Intergovernmental organizations. (CC-BY-IGO). https://data.humdata.org/faqs/licenses.

uptake of the interventions. In parallel, the NMCP provided diagnosis (HRP2-based RDT and microscopy) and treatment (artemether–lumefantrine as first line drug for clinical cases) to patients presenting at health facilities or community health workers. To monitor malaria incidence and prevalence, the Magude project established a rapid case reporting system and conducted annual cross-sectional malaria prevalence surveys [9,12]. The epidemiological results of the project, including information on the MDA campaigns, have been published elsewhere [12], as well as detailed information on the implementation and coverage of IRS [14], access and use of LLINs [15], the susceptibility of the main local vector species to insecticides [14] and a detailed evaluation of the overlap between human and mosquito behaviors [16]. This paper combines novel and detailed data on mosquito bionomics with those data that are analyzed previously to improve our understanding of the effectiveness of the project's approach.

## Entomological surveillance design

Mosquitoes were collected monthly between May 2015 and October 2017 in six sentinel sites in Magude district (Fig 1): Magude Sede, Muginge, Panjane, Mapulanguene, Chicutso and Motaze (Fig 2). These six sentinel sites were selected to represent the range of environmental and land use characteristics of the district.

In each sentinel site, mosquitoes were collected in fifteen households during two consecutive nights. They were collected indoors in 10 households and outdoors in another 5 households using CDC miniature light-trap (Model 512, John W Hock, Florida, USA). A Collection Bottle Rotator (Model 1512, John W Hock, Florida, USA) was added to six traps (three indoors and three outdoors, every night) to assess the time of mosquito host-seeking activity. The same houses were visited every month, but each month they were randomly assigned a trap type (i.e. CDC-light trap with or without a rotator) and a collection location (indoor or outdoor). Indoors, the CDC light-trap was hung at the foot-end of a bed with the trap opening approx. 1.5m above the ground. One or two adult (>15 years old) volunteers from the selected household were asked to sleep in the bed under an LLIN during the night. Participants not owning a net were provided with a WHO-pre-qualified pyrethroid-only LLIN. Outdoors, CDC light

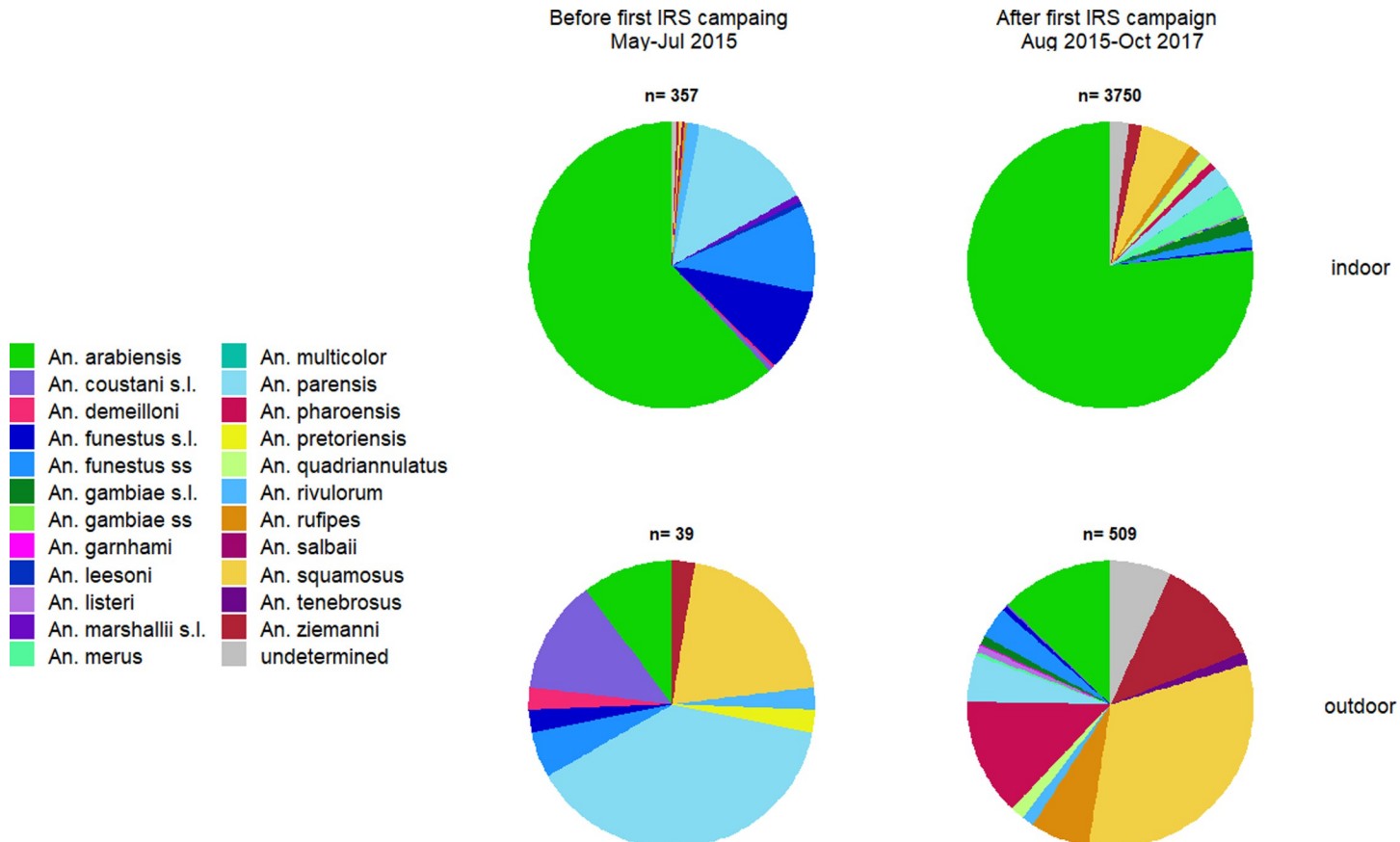

**Fig 2. Indoor and outdoor anopheline species composition before and after the implementation of the first IRS campaign during the Magude project.**

traps were baited with a BG-Lure cartridge (Biogents AG, Germany) and $CO_2$ (generated through a mixture of 10g commercially available yeast (Instant yeast, Smart Chef), 100g white refined household sugar and 1L of regular tap water) to mimic indoor conditions (i.e. a human sleeping next to the trap). The outdoor traps were placed in the safest possible outdoor environment: under a tree close to the house, but away from animals and children. Due to suspicion of arboviral disease transmission in Mozambique, which was later confirmed [17,18], no comparisons against human landing catches (HLC) were done. Hence, we discuss here exposure to host-seeking mosquitoes rather than providing human biting rates.

Every morning after a collection night, the team visited the house to retrieve the collected mosquitoes and to record information on the quality of the collection using a digital structured questionnaire. This served to exclude collections that did not match our inclusion criteria (listed in S1 File). Data were collected with tablets (Huawei, Model S7-701u) using Open Data Kit. The collected mosquitoes were taken to the laboratory. *Anopheles* mosquitoes were selected and identified morphologically to species using a stereomicroscope and the keys of Gillies and Coetzee [19]. Individuals belonging to the *An. gambiae* and *An. funestus* species complex were transferred to the lab and identified to species level using the polymerase chain reaction (PCR) [20,21]. The presence of *P. falciparum* sporozoites in individual mosquito samples was analyzed through screening enzyme-linked immunosorbent assays (ELISA) conducted on mosquitoes' grinded head and thorax [22]. The presence of sporozoites of other *Plasmodium* species was not tested because *P. falciparum* is known to account for over 90% of

all diagnosed malaria infections in Mozambique [23] and for almost all in the neighboring district of Manhiça [24], and because very low positivity rates were expected given the elimination context. Positive samples were confirmed through a second ELISA test. ELISA lysates were not heated before running the test and positive samples were not confirmed by PCR or gene sequencing.

**Indoor resting vectors.** Indoor resting mosquitoes were not collected systematically, however, indoor resting blood-fed mosquitoes were collected for insecticide resistance monitoring purposes. Mosquitoes were collected from 6 am to 10 am using a mouth aspirator and a torch from April to September and in December of 2015, from February to August of 2016 and from August to November of 2017. A descriptive analysis is provided, as indoor resting behaviors are closely linked with the success of IRS campaigns (i.e. IRS products kill susceptible mosquitoes that rest on sprayed surfaces indoors).

## Data analysis

The analysis aimed to 1) evaluate anopheline composition and densities over time; 2) quantify *P. falciparum* sporozoites per mosquito species and the impact of interventions on those sporozoite rates over time; 3) evaluate the association of different vector species with reported malaria cases, 4) evaluate the location and time of vector host-seeking activity and 5) identify the vector species that rested indoors. With these aims, the analysis makes use of the data collected through the surveillance system described in detail above as well of previously published datasets on malaria incidence, the efficacy of IRS, on ITN and MDA coverage and climatic data [12]. The results are discussed considering results from previously published analyses (see section below) to better understand the impact of the implemented interventions on local vector populations. Since the Magude project did not have a control area, nor a sufficiently long entomological baseline, this examination is mostly qualitative in nature, although we tried to quantify the impact where possible (e.g. the impact of interventions on sporozoite rates).

**Evaluation of *Anopheles* composition and densities over time.** We first evaluate *Anopheles* species compositions by calculating the relative abundance of each *Anopheles* species (i.e. the proportion of mosquitoes that belonged to each *Anopheles* species out of the total number of anophelines collected). We calculate each species' relative abundance for the period before the first intervention of the project (May to July 2015, before the first IRS campaign), and for the full intervention period (August 2015 to October 2017). For the intervention period, we calculate the relative abundance of each species indoors and outdoors separately. We then calculate the ratio of mosquitoes collected indoors to those collected outdoors to evaluate the overall endophagy of each species during the intervention period. Since the number of collections indoors and outdoors per month was different, we normalized the number of mosquitoes collected indoors and outdoors in each month by dividing those numbers by the number of collections conducted indoors and outdoors, respectively, that month.

We calculate the number of host-seeking *Anopheles* mosquitoes per person per month as the mean value of the number of host-seeking *Anopheles* mosquitoes per person in each collection within that month, separated by species. The number of host-seeking *Anopheles* mosquitoes per person in each collection was collected as the number of *Anopheles* collected divided by the number of people who slept under a net next to the trap in the collection room that night. For outdoor collections, we assumed that our artificial lure mimicked a single person, and hence the number of mosquitoes that was collected is divided by 1. The number of monthly host-seeking mosquitoes per person for each *Anopheles* species from May 2015 until October 2017 is plotted alongside intervention coverage, use and/or efficacy, malaria incidence, as well as temperature and rainfall data, with the aim to visually explore potential

associations between vector bionomics, interventions, climate and malaria incidence. Interventions coverage, use and efficacy data were obtained from previous publications [12,14,15]. Rainfall data were obtained from the Climate Hazards Group InfraRed Precipitation with Station data (CHIRPS). Data from every raster file per month were extracted for every household in Magude and aggregated to obtain monthly representative values [25]. Temperature data was obtained from the National Oceanic and Atmospheric Administration (NOAA) collected by the Maputo Weather Station (station ID 673410).

***P. falciparum* sporozoite rates and impact of interventions on these rates over time.** Since very few mosquitoes were found carrying sporozoites throughout the entire project, we present the overall sporozoite rate (i.e. the number of *P. falciparum* positive mosquitoes over all mosquitoes analyzed) and the number of *Anopheles* mosquitoes of each species that were found carrying *P. falciparum* sporozoites, separated by those that were collected indoors or outdoors. Overall sporozoite rates are subsequently provided for specific project periods related to the time of implementation of MDA and IRS, with the aim to understand the potential impact of these intervention on sporozoite rates. The impact of both MDA and IRS on sporozoite rates can already be observed two to three weeks after implementation. MDA with DHAp immediately eliminates gametocides from humans, which prevents feeding mosquitoes from ingesting gametocytes and becoming infective. The time between gametocyte ingestion and sporozoite migration to mosquito salivary glands can be two or three weeks depending on temperature. By then, a proportion of older infected mosquitoes will have died, only a few younger mosquitoes will be infected and hence sporozoite rates will be lower than before MDA. IRS immediately reduces vector densities through mortality-inducing effects, reducing parasite transmission success (from human to mosquitoes and vice versa). In addition, new mosquitoes that emerge during the two or three weeks after each IRS round are unlikely to become infected due to the lower levels of circulating parasites in the human population. As a result, sporozoite rates are expected to decrease. Since the temporal resolution of our data is monthly, we considered the following periods for sporozoite analysis to assess how both interventions may have impacted transmission: 1) prior to the first IRS campaign (May-July 2015), 2) between the start of the first IRS and the start of MDA 1 (August–October 2015), 3) during MDAs 1 and 2 (November 2015-February 2016), 4) at the end the high transmission season 2016 (March-July 2016), 5) between the start of second IRS and the start of third MDA (August to November 2016), 6) During MDAs 3 and 4 (December 2016 to March 2017), 7) at the end of the high transmission season 2017 (April to July 2017) and 8) from August to October 2017. Sporozoite rates are presented together with their 95% confidence intervals (CIs), calculated as confidence intervals of a population proportion assuming the sample meets the Central Limit Theorem (Table 1). Because we had low numbers of mosquitoes and low vector sporozoite rates, no further statistical analyses were undertaken.

**Association between vector species densities and malaria incidence.**  To understand the relative importance of each potential vector in malaria transmission and given the fact that the number of sporozoite positive mosquitoes were too low to obtain accurate estimates of the entomological inoculation rates, the association between the number of host-seeking anophelines of each species collected per month per person and the monthly malaria incidence was explored through a negative binomial multivariate regression model. The results of this model are later combined with data on the relative vector abundance, vector behaviors and how those overlap with human behaviors, to understand which species were most likely the main malaria vectors during the project.

The model correlated monthly malaria cases, as diagnosed by RDTs, with the monthly number of mosquitoes collected per person per month of all *Anopheles* species that represented at least 1% of the total vector population. The analysis was not restricted to those

**Table 1. *Plasmodium falciparum* sporozoite rates of the five *Pf* positive malaria vector species during the Magude project at distinct relevant periods defined in relation to interventions.** The proportions with 95% CI, and number of positive with respect to total mosquitoes collected in the period are shown.

| Species | Before first IRS campaign (May-Jul 2015) | Between the start of the first IRS and the start of MDA 1 (Aug-Oct 2015) | During MDAs 1 and 2 (Nov-Feb 2016) | End the high transmission season 2016 (Mar-Jul 2016) | Between the start of second IRS and the start of third MDA (Aug-Nov 2016) | During MDAs 3 and 4 (Dec-Mar 2017) | End the high transmission season 2017 (Apr-Jul 2017) | From Aug 2017 to October 2017 |
|---|---|---|---|---|---|---|---|---|
| *An. arabiensis* | 5.4% [3–9.5] (12/222) | 2.9% [1.1.-6.9] (5/174) | 0.0% [0–2.8] (0/168) | 0.2% [0–0.7] (2/1165) | 0% [0–4.2] (0/109) | 0.3% [0.1–1.2] (2/655) | 1.3% [0.6–2.9] (27/525) | 11.8% [3.8–28.4] (4/34) |
| *An. funestus s.s* | 2.7% [0.1–15.8] (1/37) | 0% [0–37.1] (0/9) | Sporozoite presence not analyzed (n = 3) | 0.0% [0–28.3] (0/13) | 0% [0–60.4] (0/4) | 0.0% [0–37.1] (0/9) | 0.0% [0–17.2] (0/24) | 0.0% [0–53.7] (0/5) |
| *An. parensis* | 1.6% [0.1–9.7] (1/63) | 0% [0–12.6] (0/34) | No *An. parensis* collected | No *An. parensis* collected | 0% [0–94.5] (0/1) | No *An. parensis* collected | 0% [0–69] (0/3) | 0.0% [0–34.5] (0/10) |
| *An. squamosus* | 0% [0–40.2] (0/8) | 20% [1.1–70.1] (1/5) | 0% [0–80.2] (0/2) | 0% [0–80.2] (0/2) | No *An. squamosus* collected | Sporozoite presence not analyzed (n = 235) | Sporozoite presence not analyzed (n = 121) | Sporozoite presence not analyzed (n = 14) |
| *An. merus* | No *An. merus* collected | No *An. merus* collected | No *An. merus* collected | 0.% [0–9.4] (0/47) | 0.0% [0–94.5] (0/1) | 0.0% [0–12.6] (0/36) | 0.0% [0–10.7] (0/41) | 20% [1.1–70.1] (1/5) |
| All five species combined | 3.7% [2.1–6.2] (14/330) | 2.5% [1–5.6] (6/222) | 0% [0–2.5] (0/186) | 0.2% [0–0.6] (2/1295) | 0.0% [0–3.9] (0/120) | 0.3% [0–1.1] (2/757) | 1.3.% [0.6–2.7] (0/611) | 7.5% [2.8–17.3] (5/67) |

vectors carrying sporozoites, as other anopheline species collected during the project are known vectors in surrounding countries [26], and the low number of mosquitoes collected of some of these species may have prevented us from detecting sporozoites in their population. Monthly LLIN use, IRS residual efficacy and MDA coverage were included in the analysis because these interventions confound the effect of vector densities on malaria cases, as they can prevent mosquito entry into houses, reduce contact between vectors and humans and reduce the proportion of infected vectors. Since our temporal resolution is a month but the effect of the implemented interventions on malaria cases can be observed within two-three weeks (given the biological cycle of malaria transmission and the vector's lifecycle) two models were fitted: one considering unlagged covariates and one considering covariates lagged one month. In addition, we fitted these two models for the entire vector surveillance period from May 2015 to October 2017 (i.e. including the baseline period May to July 2015), but also for the period August 2015 to October 2017 (intervention period only). The number of visits to a health facility was used as the offset, to account for variations in care seeking behaviors over time. Malaria incidence data were obtained from the DHIS2-based rapid case reporting system established in Magude district in January 2015 [12]. MDA coverage was assumed to be equal to the campaign coverage during the two months that each campaign lasted and 0 for the other months. MDA coverage estimates were obtained from previously published analyses [12]. Monthly IRS efficacy estimates for the 2016 and 2017 campaigns were obtained by fitting a logistic binomial Bayesian model to the observed mosquito mortality data 24h-post exposure in WHO standard cones bioassays. More details on the collection of residual efficacy data, the residual efficacy data themselves, and the data analysis are provided elsewhere [14]. Since the residual efficacy of the products used during the 2015 IRS campaign was not monitored, we assumed that the residual efficacy of Actellic 300CS and DDT in 2015 was similar to the efficacy of Actellic 300CS alone in 2016. Similar residual efficacies for both products have been observed in other campaigns [27], and due to the overall high variability of the residual efficacy

of IRS products across geographical locations [27–29], we refrained from including DDT residual efficacy data from other sites. To calculate the residual efficacy of the 2015 IRS campaign, the residual efficacy of Actellic 300CS in 2015 was adjusted for the observed pace of spraying and coverage (94.5%) of the campaign that year (following the exact same method described elsewhere for the 2016 and 2017 IRS campaigns [14]. For the models that included the baseline period (May-July 2015), the residual efficacy of the 2014 IRS with deltamethrin was considered zero, as only Motaze (that accounts for 13.5% of the Magude population) received IRS and because the optimal residual efficacy of deltamethrin has been observed to be between 3–6 months in other settings and is therefore expected to have waned by May 2015 when the project started [27]. LLIN use was measured several times during the Magude project (details on the methodology and data collection are provided elsewhere [15] and followed a seasonal pattern, which was modelled using a sinusoidal function,

$$f(x) = A \sin B(x - C) + D$$

where x is the month, A is the amplitude of the variation which we modeled as $amplitute = \frac{\max (ITN_{observed\ use}) - \min (ITN_{observed\ use})}{2}$, B is the period, which for months is $\frac{2\pi}{12}$, C was adjusted for the sinusoidal function to follow the seasonality of LLIN use and D is the minimum observed use (39.1%) plus the amplitude of the variation (the function is represented in S2 File). The goodness of fit was evaluated by checking the distribution and autocorrelation of the residuals. Models were compared using the Akaike Information Criterion. The model with the lowest AIC was considered to be the best performing model, provided there was no autocorrelation in its residuals. Regression log transformed coefficients are reported together with their 95% confidence intervals. The predicted cases are shown along with true malaria cases. Detailed model results are provided in S4 File.

**Vector host-seeking activity.** We evaluated indoor and outdoor host-seeking times during the project's intervention period (August 2015 to October 2017) by calculating the number of host-seeking mosquitoes of each species collected per person for each 2 hour time interval (period of rotation of the CDC bottle rotator) from 18:00 to 06:00, before 18:00 and after 06:00, separating indoor and outdoor collected mosquitoes. Then we evaluated the composition of host-seeking vectors at the different collection time intervals by calculating the relative percentage of the total host-seeking mosquitoes per person that each species represented.

All data cleaning and analysis was conducted using R version 4.1.0.

## Ethical clearance

Ethical approval was obtained from the Manhiça Health Research Center's Institutional Bioethics Committee for Health (CIBS-CISM/043/2015 for our entomological surveillance) and local administrative authorities (52/SDSMASS/024.1). Verbal informed consent was obtained from an adult member of each household where a mosquito trap was placed indoors or outdoors. All participating households were free to withdraw from the studies at any given time. All other studies were approved by CISM's institutional ethics committee, Hospital Clinic of Barcelona's Ethics Committee, and the Mozambican Ministry of Health National Bioethics Committee. The study protocol to implement and evaluate the impact of MDAs was also approved by the pharmaceutical department of the Ministry of Health of Mozambique and registered as Clinical Trial NCT02914145. More details on the ethical considerations of the population census, household surveys, cross-sectional prevalence surveys and MDAs are provided elsewhere [12,13]

## Results

### Mosquito collections

A total of 5,361 trap-night collections were performed between May 2015 and October 2017. Of those, 513 collections were discarded for not complying with the inclusion criteria (S1 File). As a result, 4,848 trap-night collections were considered in the present analysis, 3,329 indoors (933 in CDC light traps with collection bottle rotators, and 2,396 in CDC light traps without rotators) and 1,519 outdoors (808 in CDC light traps with collection bottle rotators, and 711 in CDC light traps without). Only 18.4% of the trap-night collections yielded at least one female *Anopheles* mosquito, with 81.6% resulting in zero mosquitoes caught. A total of 4,655 *Anopheles* female mosquitoes were caught, 4,107 indoors (1,015 in CDC light traps with bottle rotators and 3,092 with CDC light traps without) and 548 outdoors (243 with CDC light traps with bottle rotators and 305 in CDC light traps without). Accounting for the differences in the number of sampling nights indoors and outdoors, these numbers indicate indoor-outdoor ratios of roughly 3.4 to 1 respectively.

### Anopheline species composition and densities over time

Ninety-seven percent (97.5%, n = 4,539) of all collected mosquitoes were identified morphologically. Of the indoor collected mosquitoes 1.9% (n = 93) could not be identified; of the outdoor collected mosquitoes 5.9% (n = 37) could not be identified because they were either too damaged or because the microscopists could not find a matching species in the dichotomous keys. Molecular identification was performed for 98% (n = 3,364) of mosquitoes belonging to the *An. gambiae* complex, and 87.3% (n = 332) of mosquitoes belonging to the *An. funestus* group.

Before the scale up of IRS (May-July 2015, i.e. the dry season), mosquitoes from the *An. gambiae* complex (all identified as *An. arabiensis*) accounted for 56.8% (n = 225) of the *Anopheles* collected, and those from the *An. funestus* group accounted for 36.1% (n = 143). Most mosquitoes from the latter species group were identified as *An. parensis* (44.8%, n = 64), followed by *An. funestus s.s.* (23.8%, n = 37), *An. leesoni* (1.4%, n = 2) and *An. rivulorum* (4.2%, n = 6). The other 23.8% (n = 34) of the mosquitoes in this group could not be identified to species. *An. squamosus* accounted for 2.3% (n = 9) of the anopheline collected. Other *Anopheles* species accounted for 6.8% of the mosquito population, but less than six individuals of each of these other species were collected.

From the scale up of IRS (August 2015) onwards (i.e. intervention period), mosquitoes belonging to the *An. gambiae* complex continued to account for the majority of anophelines collected (75.3%, n = 3,206). Molecular identification revealed the following composition: *An. arabiensis* (91.6%, n = 2,938), *An. merus* (4.2%, n = 135), *An. quadriannulatus* (2%, n = 63), *An. gambiae s.s.* (0.1%, n = 3) and 2.1% (n = 67) could not be identified. *An. squamosus* accounted for 8.9% (n = 380) of the *Anopheles* collected followed by the *An. funestus* group (5.6%, n = 237). The species composition within this group was *An. parensis* (50.2%, n = 119), *An. funestus s.s.* (37.6%, n = 89), *An. rivulorum* (5.1%, n = 12), *An. leesoni* (0.8%, n = 2) and 6.5% (n = 15) could not be identified. Finally, *An. ziemanni* accounted for 2.7% (n = 114), *An. pharoensis* for 2.3% (n = 99), *An. rufipes* for 1.9% (n = 83) and several other vector species for 3.3%, with less than 12 mosquitoes of each of these other species collected.

The indoor vector composition during the intervention period was: *An. arabiensis* (76.6%, n = 2,873), *An. squamosus* (5.8%, n = 217), *An. merus* (3.5%, n = 133), *An. parensis* (2.5%, n = 92), *An. funestus s.s.* (1.9%, n = 71), unidentified mosquitoes of the *An. gambiae* complex (1.7%, n = 62), *An. quadriannulatus* (1.5%, n = 55), *An. ziemanni* (1.4%, n = 52), *An. rufipes*

(1.3%, n = 49), *An. pharoensis* (0.9%, n = 32), unidentified mosquitoes of the *An. funestus* group (0.3%, n = 12), *An. listeri* (0.2%, n = 6), *An rivulorum* (0.1%, n = 5), *An. coustani s.l.* (0.1%, n = 4), *An. gambiae s.s.* (0.1%, n = 3), *An. leesoni* (0.1%, n = 2), *An. tenebrosus* (0.1%, n = 2), *An. multicolor* (0%, n = 1) and 2.1% (n = 79) could not be identified (Fig 2).

The outdoor vector composition during the intervention period was: *An. squamosus* (32%, n = 163), *An. pharoensis* (13.2%, n = 67), *An. arabiensis* (12.8%, n = 65), *An. ziemanni* (12.2%, n = 62), *An. rufipes* (6.7%, n = 34), *An. parensis* (5.3%, n = 27), *An. funestus s.s.* (3.5%, n = 18), *An. quadriannulatus* (1.6%, n = 8), *An. rivulorum* (1.4%, n = 7), *An. tenebrosus* (1.4%, n = 7), unidentified mosquitoes of the *An. gambiae* complex (1.0%, n = 5), *An. listeri* (0.8%, n = 4), unidentified mosquitoes of the *An. funestus* group (0.6%, n = 3), *An. merus* (0.4%, n = 2), *An. coustani s.l.* (0.2%, n = 1), *An. garnhami* (0.2%, n = 1) and 6.9% (n = 34) could not be identified (Fig 2).

After normalizing the total number of mosquitoes collected indoors and outdoors by the number of collection nights indoors and outdoors, respectively, the ratio indoor to outdoor collected mosquitoes for the different species during the intervention period was as follows: *An. arabiensis* (20.0 indoors to 1 outdoors), *An. merus* (30.1:1), *An. quandriannulatus* (3.1:1), *An. coustani s.l.* (1.8:1), *An. funestus s.s.* (1.8:1), *An. parensis* (1.5:1), *An. listeri* (0.7:1), *An. rufipes* (0.7:1), *An. squamosus* (0.6:1), *An. ziemanni* (0.4: 1), *An. rivulorum* (0.3:1), *An. pharoensis* (0.2: 1) and *An. tenebrosus* (0.1:1). This suggests that *An. arabiensis* and *An. merus* were highly endophagic, that *An. coustani s.l.*, *An. funestus s.s.*, and *An. parensis* were slightly more endophagic than exophagic and that *An. listeri*, *An. rufipes*, *An. squamosus*, *An. ziemanni*, *An. rivulorum*, *An. pharoensis* and *An. tenebrosus* were more exophagic.

The number of indoor host-seeking mosquitoes per person oscillated seasonally reaching a maximum in the month of April in both years, after the annual rainfall peak. More than 55% of the total host-seeking anopheline mosquitoes per person were collected during the months of February, March and April, with 27.7% of all mosquitoes being collected during the month of April (Fig 3). The greatest species richness was observed during the first half of 2017 (Fig 3), following the highest rainfall, and preceding the highest malaria incidence observed during the project.

Indoors, *An. arabiensis* was the predominant vector before and during the intervention period, accounting for between 48% and 100% of all host-seeking anophelines collected per person in any given month until September 2017. The proportion of host-seeking *An. funestus s.s.* and *An. parensis* decreased markedly after the first IRS campaign. *An. parensis* disappeared and was only collected again in November 2016 and from July 2017 onwards, accounting for 62.2% and 27.0% of host-seeking anophelines in September and October 2017, respectively. *Anopheles funestus s.s.* was occasionally present during the period when IRS was effective, but only accounted for 0.6% up to 12.1% of indoor host-seeking anophelines when it was found. *An. merus* was only collected after the first IRS campaign and accounted for between 1.2% and 26.0% of host-seeking anophelines when it was found. Indoor host-seeking *An. squamosus* accounted for between 0.2% and 4.0% of host-seeking anophelines before and after the first IRS campaign until February 2017, when it accounted for 17.5%. Its relative presence remained high in subsequent months (14.5% and 14.9% in March and April 2017, respectively). *An. ziemanni* was only present in larger numbers in 2017, but still accounted for less than 5% of the vectors collected indoors until May and June 2017, when its relative abundance increased to 7.2% and 7.8%, respectively (Fig 3).

Outdoors, different species dominated in different months. There was no single species that predominated over time during the intervention period, and several species were only collected outdoors in specific months. Two peaks in the number of outdoor host-seeking anophelines per person were observed. The first peak was in April 2016, the second in February 2017,

Number of mosquitoes collected and malaria cases

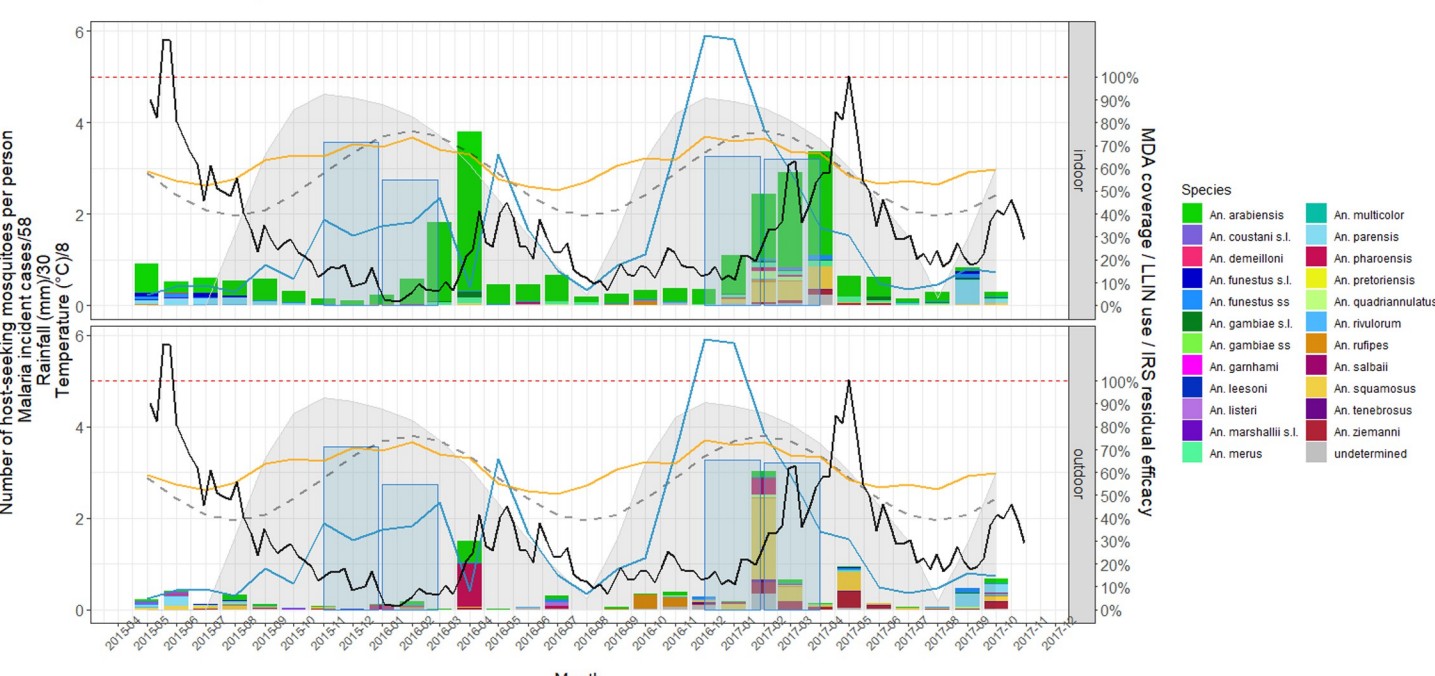

**Fig 3. Anopheline densities over time during the Magude project in relation to malaria cases, climate data, and relevant malaria control intervention indicators.**
Number of anopheline mosquitoes collected (per person per month, colored bars), malaria cases (black line), modeled LLIN use over time (%, dashed grey line), modeled IRS efficacy over time (%, grey shaded areas), MDA coverage (%, blue shaded areas), amount of rainfall (mm, blue line) and temperature (˚C, orange line).

with 15.1% and 31.0% of all mosquitoes collected after the first IRS campaign, respectively. The peak in April 2016 was dominated by *An. pharoensis* and preceded an increase in malaria cases. The peak in February 2017 was dominated by host-seeking *An. squamous* and was also followed by an increase in malaria cases. Outdoor host-seeking *An. arabiensis* were collected outdoors throughout the project. Outdoor host-seeking *An. parensis* accounted for a high proportion of outdoor host-seeking anophelines before the first IRS campaign (17.6% to 55.5%) and were absent until September and October 2017, when they accounted for 61.9% and 27.8%, respectively. Outdoor host-seeking *An. funestus s.s.* were sporadically collected throughout the study. Outdoor host-seeking *An. squamosus* accounted for between 15.4% and 37.5% of all outdoor host-seeking *Anopheles* before the first IRS campaign, but were mostly absent until December 2016. Between December 2016 and July 2017, they accounted for between 18.1% and 66.6% of the monthly outdoor host-seeking *Anopheles* collected. *An. ziemanni* was absent from outdoor collections from the first implementation of IRS until December 2016, when its relative abundance increased up to 42.8% in June 2017, preceding an increase in malaria incidence, and remained substantial until the end of the implementation period (October 2017) (Fig 3).

## Sporozoite rates

The presence/absence of *Plasmodium falciparum* sporozoites was determined for 3,656 specimens (78.5% of all mosquitoes collected). A total of 37 (0.8%) mosquitoes were sporozoite positive; 35 collected indoors and two collected outdoors. These belonged to five species: *An. arabiensis* (32 positive of 3,052 tested, 1%), *An. merus* (1/128, 0.8%), *An. parensis* (1/111, 0.9%), *An. funestus s.s.* (1/101, 1%), *An. squamosus* (1/17, 5.9%), and one unidentified

mosquito from the *An. gambiae* complex (1/62, 1.6%). Of the 59 *An. quadriannulatus*, 46 *An. pharoensis*, 15 *An. rivulorum*, 8 *An. rufipes*, 6 *An. coustani s.l.*, 5 *An. ziemanni*, 3 *An. gambiae s. s.*, 3 *An. marshallii* complex, 2 *An. leesoni*, 1 *An. demeilloni*, 1 *An. garnhami*, 1 *An. pretoriensis* and 1 *An. tenebrosus* analyzed, none were positive. None of the *An. listeria*, *An. multicolor* or *An. salbaii* were tested for sporozoites.

Indoors, sporozoite rates were 1% for *An. arabiensis* (31/2987), 0.8% for *An. merus* (1/126), 1.1% for *An. parensis* (1/87) and *An. funestus s.s.* (1/88) and 33.3% for *An. squamosus* (1/3). The two outdoor sporozoite positive mosquitoes included one *An. arabiensis* (1/65, 1.5%) and one mosquito from the *An. gambiae* complex that could not be identified to species (1/5, 20%).

Sporozoite-positive *An. arabiensis* were detected for the duration of the project while other species only tested positive sporadically. After the first MDA campaign, only *An. arabiensis* and *An. merus* were found positive for *P. falciparum*. The sporozoite rates for the five vector species during specific points in time (described in the methods and related to the timing of our malaria and vector control interventions) are shown in Table 1. More details on sporozoite rates per species and month are provide in S3 File.

Overall, vector sporozoite rates decreased with the implementation of IRS and the two first rounds of MDA but increased again at the end of the high transmission season of 2017 when the highest malaria incidence of the entire project was observed (Table 1). An unexpected increase in sporozoite rates was subsequently observed in August 2017 (12.5% [5.2–25.9]) and September 2017 (20.0% [6.6–44.3]) (S3 File) and only *An. arabiensis* and one *An. merus* mosquito were sporozoite positive during those months.

## Association between host-seeking *Anopheles* per person and residual malaria incidence

Only species accounting for more than 1% of the anophelines collected were included in the analysis. Of the models used to correlate the number of host-seeking mosquitoes per person per month and per species, and controlling for MDA coverage, LLIN use and IRS efficacy, the best-performing model was the one including covariates lagged one month and which only considered the intervention period (August 2015 to October 2017). It had an AIC of 318.55 compared to 348.4, 377.2, 399.4 for the other models (detailed results from each model are presented in S4 File). In this best performing model, *An. arabiensis*, *An. funestus s.s. An. parensis*, *An. squamosus*, *An. merus*, *An. rufipes* and *An. quadriannulatus* were significantly associated with malaria incidence. *An. arabiensis* at <1% significance, *An. parensis* and *An. quamosus* at 1% significance, *An. funestus s.s.*, *An. merus* and *An. quadriannulatus* at 5% significance and *An. rufipes* at 10% significance. The coefficients, reflecting the likely increase in malaria cases expected with the respective presence of each mosquito if positive, were: *An. arabiensis* 2.93 [95% CI: 2.06–4.19], *An. parensis* 23.4 [4.41–248.58], *An. squamosus* 30.21 [4.21–218.55], *An. funestus s.s.* $6.76\times10^{-6}$ [$3.64\times10^{-10}$–0.11], *An. merus* $2.75\times10^{-2}$ [$1.14\times10^{-3}$–0.74], *An. rufipes* 0.04 [$1.42\times10{-3}$–1.26], *An. quadriannulatus* $4.31\times10^{-7}$ [$2.7\times10^{-12}$–0.08]. Fig 4 shows the actual reported malaria cases and the cases predicted by the best performing model.

## Place and time of host-seeking anopheline mosquitoes

Fig 5 shows both the composition of anopheline host-seeking mosquitoes and their densities for each 2h interval during the evening, night and early morning during the intervention period (August 2015 to October 2017). Indoors, most of the host seeking mosquitoes were *An. arabiensis*. Host seeking activity was concentrated in the evening and nighttime hours before 02:00 and showed the greatest peak between 00:00 and 02:00. Outdoors, host-seeking activity was concentrated between 18:00 and 00:00 and between 02:00 and 04:00. Early hours saw a

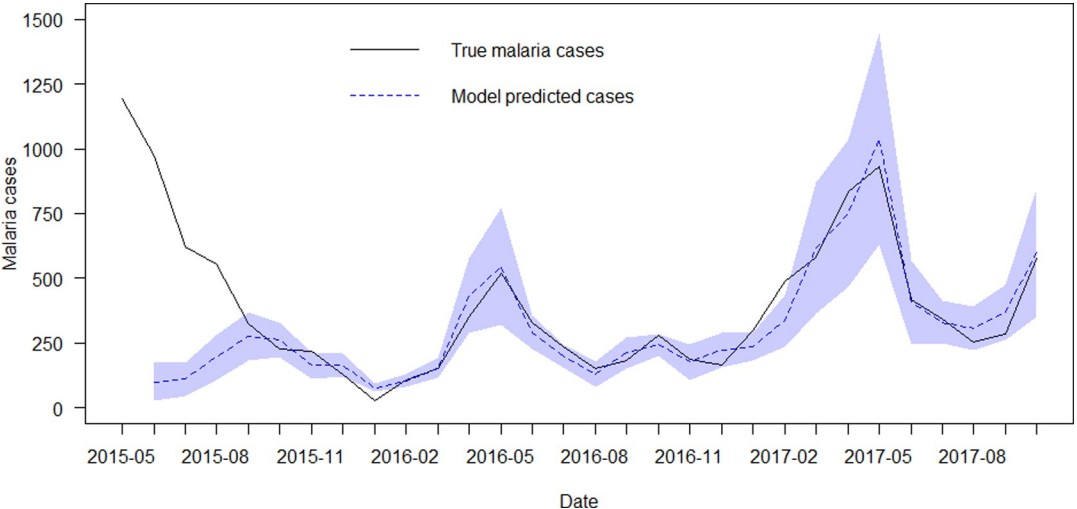

**Fig 4. True malaria cases versus model predicted cases.** Shaded areas represent 95% CI.

marked presence of host-seeking mosquitoes of the *An. funestus* group and *An. squamosus* mosquitoes. The peak between 02:00 to 04:00 was dominated by *An. squamosus* and *An. pharoensis*.

## Identification of indoor resting vectors

A total of 1,042 blood-fed *An. funestus s.l.* mosquitoes were collected indoors in 2015, before the first IRS campaign was implemented. Very few mosquitoes of this group (insufficient for

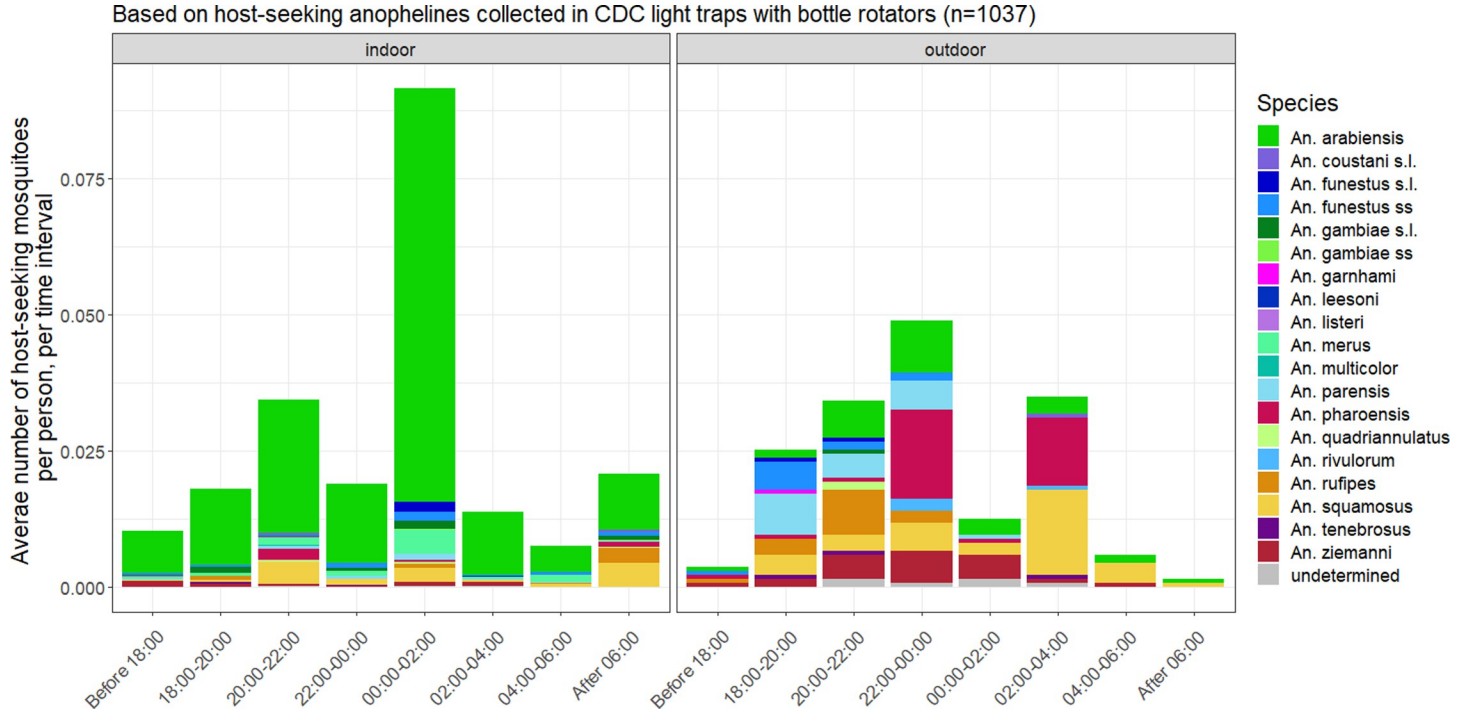

**Fig 5. Density of host-seeking mosquitoes over different time intervals during the evening, night and early morning.**

insecticide susceptibility testing) were collected indoors during the years following the first IRS campaign. Blood-fed mosquitoes of the *An. gambiae* complex were found resting indoors in sufficient numbers for resistance testing during the entire intervention period (1,024 in 2015, 3,753 in 2016 and 508 in 2017). A subset of *An. funestus s.l.* mosquitoes collected indoors in 2015 were identified, and the following species detected: *An. parensis*, *An. funestus s.s.*, *An. rivolorum* and *An. vaneedeni*. Among the *An. gambiae* complex, most individuals were *An. arabiensis*, a few were *An. merus*. Although indoor resting behavior was not assessed through pyrethrum spray catches or similar methods, these data suggest that a part of the population of members of the *An. funestus* group and of *An. arabiensis* may rest indoors during a part of their gonotrophic cycle.

## Discussion

The present study aimed to i) identify the anopheline species that sustained malaria transmission during the Magude project, ii) qualitatively evaluate the impact of the interventions on those species, and iii) identify the potential gaps in vector control during the project, to provide recommendations for future malaria elimination efforts in the area.

We believe that five species sustained transmission during the Magude project—*An. arabiensis*, *An. merus*, *An. funestus s.s.*, *An. parensis* and *An. squamosus*—with others potentially playing a very minor role, if any. These five species were found positive for *P. falciparum* sporozoites and were significantly associated with malaria incidence. In southern Mozambique, *An. funestus s.s.* and *An. arabiensis* have historically been identified as the major malaria vectors [7,8,30]. *Anopheles merus* was first incriminated as a vector in year 2000 in Boane, located in Maputo province [31] and later during the Lubombo Spatial Development Initiative in the same province [8]. In contrast, and to our best knowledge, this is the first time that *An. squamosus* and *An. parensis* are found carrying *P. falciparum* sporozoites in southern Mozambique. *Anopheles squamosus* has been incriminated as a malaria vector in southern Zambia, but its exact role in malaria transmission in that area is unknown [32]. In 2017, *An. parensis* was identified as a minor vector species in Kwazulu-Natal, a province in South Africa that borders the southern part of Mozambique [33,34]. Although *P. falciparum* sporozoites were not detected in any of the *An. rufipes* (8 analyzed out of 83 identified) and *An. quadriannulatus* (59 analyzed out of 63 identified), these two species were significantly associated with malaria cases, albeit with the weakest associations and very small correlation coefficients. Although it has been demonstrated that *An. quadriannulatus* can carry *P. falciparum* sporozoites in laboratory studies [35], this species has never been incriminated as a malaria vector in nature and therefore unlikely played a role in malaria transmission during the Magude project. *Anopheles rufipes*, on the other hand, is a secondary malaria vector in equatorial countries of Africa (Burkina Faso, Cameroon, Gambia, Ghana, Kenya, Mali, Nigeria, Senegal and Togo) as well as in Zambia [26,36–44]. As such, this vector could have played a minor role during the project.

Other anopheline species that were found in Magude district are known vectors of malaria in southern Mozambique, i.e. *An. tenebrosus* [45], or elsewhere in Africa, namely *An. coustani s.l* [46–49], *An. ziemanni* [50–53], *An. rivulorum* [49,54,55], *An. leesoni* [44,56–58] and *An. pharoensis* [39,49,59–61]. However, none of the few analyzed specimens of these species were found carrying *P. falciparum* sporozoites. In addition, several non-vector *Anopheles* species were collected, namely *An. demeilloni*, *An. garnhami*, *An. listeri*, *An. marshalli s.l.*, *An. multicolor* and *An. salbaii*. To our knowledge, this is the first time that *An. garnhami*, *An. multicolor* and *An. salbaii* are collected in Mozambique [26]. Of these, only *An. garnhami* has been detected in Mozambique's neighboring countries (i.e. South Africa, Tanzania and Zimbabwe) [26]. In contrast, *An. multicolor* and *An. salbaii* have been detected in equatorial Africa,

namely Niger and Sudan (*An. multicolor*) and in Ethiopia, Kenya, Niger, Somalia and Djibouti [26]. The detection of these other vector and non-vector *Anopheles* species could, however, be an artifact of wrong morphological identification resulting from damaged mosquito specimens [49,62]. Unfortunately, none of these species were molecularly identified to species (e.g. using ITS2 and COI analysis [49]) and the presence of sporozoites was not assessed for several *Anopheles* species. Confirming the presence of these species and investigating their role in persistent malaria transmission, especially of those known to be vectors in other countries of Africa, should be a priority in future studies in southern Mozambique.

Three specimens of *An. gambiae s.s.*, a very competent vector of malaria transmission in sub-Saharan Africa, were found in Magude district. Although this vector is nowadays rarely found in southern Africa, it was found resting indoors in Magude's neighboring district of Chokwe back in 2000–2002 [63] and it was collected in 2017–2018 in the neighboring's province of Limpopo, South Africa [64]. The detection of this vector could have been the result of misidentifications. In 2018, Erlank et al. [62] showed that *An. squamosus*, *An rufipes* and *An. pretoriensis* showed amplicons similar to *An. gambiae s.s.* when the PCR protocol for identification of species of the *An. gambiae* complex was applied to them. Hence, if mosquitoes of any of these three species were wrongly identified as belonging to the *An. gambiae* complex, this could have led into false identification of *An. gambiae s.s.* The current presence of *An. gambiae s.s.* in southern Mozambique and its potential role in transmission should be further investigated. It is recommended that if this species is reported in southern Africa in the future, the species identification be confirmed with sequencing of the ITS2 or COI regions.

Although accurate estimates of sporozoite rates or entomological inoculation rates could not be established due to i) the low numbers of mosquitoes collected, and ii) the fact that the parasite reservoir was targeted by several interventions, *An. arabiensis* was most likely responsible for most of the residual malaria transmission during the Magude project. This is because it was the most abundant mosquito species, accounted for the highest proportion of human exposure to bites [16], it presented one of the strongest statistical associations with malaria incidence and it was the only vector found carrying sporozoites after the first IRS campaign, except for one *An. merus*. *Anopheles funestus s.s.* and *An. merus* may have played a minor but continuous role, based on their significant association with malaria incidence and consistent presence through the intervention period albeit with a low relative abundance. *An. squamosus* may have played a minor but more erratic role. While it was significantly associated with malaria incidence, it was only present in the months surrounding the observed high malaria incidence peak. *An. parensis* may have played a role sporadically. Despite its significant association with malaria incidence, it disappeared after the implementation of the first IRS campaign and practically didn't appear again until July 2017. If other anopheline species contributed to residual malaria transmission, such as the two other vectors that were significantly associated with malaria incidence (*An. rufipes* and *An. quadriannulatus*) or those that are known vectors elsewhere, they likely also played a very minor role due to their very low relative abundances and densities. Evaluating the exact role of each species will be critical to guide vector control strategies but is cumbersome in a project like the Magude project due to the high pressure exerted on both vector (LLINs, IRS) and parasite populations (MDAs). The relative importance of each species should be ideally assessed in similar nearby areas that are not subjected to intense control interventions.

The project's interventions managed to significantly reduce sporozoite rates from 3.7% before the first IRS campaign, to 0.1%, after the first MDA campaign before the second IRS campaign started, but sporozoite rates later increased to 1.1% after the MDA rounds in year 2. These patterns are in line with the decrease in malaria prevalence observed through the cross-sectional prevalence surveys conducted in May 2015 and May 2016, which revealed that

malaria prevalence went from 9.1% to 1.5% after the implementation of the first IRS campaign and the two first MDA rounds, but later increased to 2.6% in May 2017 [12]. The fluctuation in sporozoite rates and malaria prevalence shows that, although the intervention package managed to reduce transmission to very low levels during the first year, it was not further reduced after the second year despite a similar coverage of all interventions. Only LLINs may have been less effective around that time (approx. two years after the mass campaign in 2014) as their integrity and insecticide bio-efficacy is known to decrease over time [65,66]. This, alongside the heavy rains in 2017, may be a reason for the slightly increased incidence rates observed that year.

IRS was likely very effective at controlling *An. parensis* and *An. funestus s.s.* because they were susceptible to the insecticides use in IRS (DDT and pirimiphos-methyl) [14] and the numbers in our CDC light trap collections decreased dramatically after the implementation of the first IRS campaign. *An. parensis* practically disappeared for approx. 2 years, whereas *An. funestus s.s.* remained present but in very low densities. In addition, very few (insufficient numbers to conduct insecticide susceptibility assays) blood-fed *An. funestus s.s.* and its sibling species were found resting indoors after the first IRS campaign, whereas mosquitoes of that species group were abundant prior to the first round of IRS. Our analysis suggests that *An. arabiensis* was not affected as much by IRS compared to *An. parensis* and *An. funestus s.s.*, even though *An. arabiensis* was also susceptible to the IRS insecticides [14]. This is because the relative abundance of *An. arabiensis* increased after IRS, it remained the predominant species throughout the project, its population managed to increase rapidly every year when rains increased and as the effect of IRS waned off, and we continued to find large enough numbers of this species indoors for our insecticide susceptibility bioassays throughout the project. Elsewhere it has been observed that *An. arabiensis* is much less affected by IRS with pirimiphos-methyl than other important vector species, such as *An. funestus s.s.* [67], possibly due to the fact that a proportion of its population rests outdoors after feeding avoiding contact with sprayed wall surfaces [68–70]. Actually, following intense implementation of IRS in southern Africa over the last eight decades, *An. arabiensis* has become the main malaria vector in South Africa and eSwatini [4,71]. Nonetheless, IRS presumably limited its population growth, as densities are expected to increase after the rains (because this vector readily breeds in small, temporary and shallow pools [72]), yet the population remained at similar densities after the heavy rains in 2017, compared to the drier preceding year. The effect of IRS on *An. merus* cannot be analyzed, as no mosquitoes of this species were collected before the first IRS campaign. However, studies conducted in Mozambique during the Global Malaria Eradication Campaign (1960–1969) showed that *An. merus* entered houses to feed, but rested outdoors, thereby avoiding contact with IRS [7]. A similar behavior is expected in Magude district, as very few *An. merus* (insufficient numbers for the insecticide susceptibility assays) were collected during indoor manual collections. The effect of IRS on *An. squamosus* cannot be properly examined due to their very low numbers, but the limited data suggest that it is likely little affected by IRS. The relative proportion of *An. squamosus* increased after the first IRS campaign and, although it maintained a very low relative abundance or was not detected during most months of the project, its population increased rapidly in February 2017, when the efficacy of IRS was waning but estimated to still be around 80%.

Looking at all species together, around 50% of mosquitoes were collected between February and April and *Anopheles* densities increased rapidly from January onwards, approximately five months after the start of each IRS campaign. This coincides with the duration of IRS's optimal residual efficacy, which was estimated to be between 3.5 months and 5.5 months, when considering mosquito mortality 24h post-exposure or delayed mortalities, respectively. It seems that IRS was effective at controlling mosquitoes during the initial months after implementation

(and during the start of the rainy season), but that its residual efficacy did not prevent the growth of vector populations during the entire malaria transmission season [14]. A second round of IRS or using a product with longer residual efficacy will be needed to effectively cover the high transmission season. However, since a second round of IRS is unlikely to be operationally feasible, given costs and the logistical challenges during a rainy season, products with a longer lasting residual efficacy will be preferred.

LLINs likely provided good protection against *An. arabiensis* (both in terms of killing susceptible mosquitoes, and reducing human-vector contact), as this species was largely endophagic, active when people were already sleeping [16] and susceptible to pyrethroids [14]. Our previous analysis showed that LLINs prevented 41.8% of human exposure to *An. arabiensis* and could have prevented 67.4% if all residents would have used a LLIN to sleep. LLINs likely provided lesser protection against *An. funestus s.s.* and *An. parensis*, compared to *An. arabiensis*. These two vector species were resistant to pyrethroids [14] and nets could only prevent 21.9% and 13.9% of exposure to host seeking mosquitoes of these species, respectively [16]. Although we could not evaluate the level of pyrethroid resistance, if any, in *An. squamosus* and *An. merus*, LLINs prevented less than half of the human exposure to these vectors (32.0% and 45.4%, respectively). As shown previously, LLIN use was suboptimal, especially during the low transmission season [15]. LLIN personal protection against host seeking mosquitoes of species that were *P. falciparum* positive during the project could have increased by between 18% to 30%, depending on the vector species, if all residents would have used the net to sleep [16]. We therefore conclude that LLINs did not achieve their full protection potential during the Magude project, but that they did complement the protection provided by IRS, mainly by providing a certain level of protection against *An. arabiensis*, *An. merus* and *An. squamosus*. Improving LLIN use will be especially important for controlling the *An. arabiensis* and *An. merus* that survive IRS, as they mainly bite indoors while people are in bed.

The results presented here, combined with previous analyses [16] show that there is a proportion of vectors that either seek their host outdoors, or seek their host indoors at times when people are not yet under a net. In addition, some species could not be found resting indoors whereas others were still found resting indoors after the implementation of IRS. This highlights that there are gaps in protection by both LLINs and IRS.

The complex changes in vector composition over time, the diversity in feeding behaviors, combined with suboptimal levels of LLIN use and short IRS realized efficacy, suggest that ITNs are IRS alone would not have been sufficient to fully control local vector populations. Additional interventions and stronger community engagement campaigns would likely have helped to achieve optimal vector control. House screening, eave tubes and lethal lures [73] could tackle those vectors that seek their hosts indoors before people are under a net. In addition, larviciding or other forms of environmental management could reduce local vector populations, including those that are not affected by current adult vector control interventions. Future interventions to kill or prevent outdoor host-seeking vectors from finding their host, including repel and lure devices and attractive targeted sugar baits, may become suitable options to increase human protection outdoors during elimination efforts in southern Mozambique.

The present study has several limitations that may have hampered our ability to fully understand the vectors that sustained malaria transmission during the project, and the impact of the interventions on those vectors. First, *P. falciparum* sporozoite rates were not assessed for all mosquito species collected, and, consequently, other species may have contributed to sustaining local malaria transmission. Second, the ELISA protocol that was followed has been observed to yield false positive results in some mosquito species, especially those that exhibit zoophilic tendencies [74]. Although we ran a confirmatory test for each positive specimen, since 1) *An.*

*arabiensis*, *An. parensis*, *An. merus* and *An. squamosus* are all known to be partially or opportunistically zoophilic and 2) the presence of sporozoites was not confirmed by PCR, some species may have been falsely identified as carrying sporozoites. Third, the low number of mosquitoes one collects during an elimination campaign due to the high pressure on the vectors (LLINs and IRS), combined with high pressure on the parasite reservoirs (MDA), hamper the possibility to reliably estimate sporozoite rates. Fourth, the human blood index was not determined for any of the vector species identified as host-seeking mosquitoes, which were mostly unfed. Understanding this is important, as the sporozoite rate combined with the preference to feed on humans and overall human biting rates determine the entomological inoculation rate, the gold standard metric to understand the relative importance of each vector species in malaria transmission. Fifth, since our CDC light traps started collecting mosquitoes several hours before people went to bed (i.e. while there was no human bait under the net next to the trap), the number of host-seeking mosquitoes per person reported for the time interval 18:00–20:00 may be an underestimation. Sixth, the residual efficacy of DDT was not measured after the 2015 campaign, which affects the accuracy of our IRS residual efficacy estimates for this campaign. Finally, due to the absence of a sufficiently long baseline of mosquito collections prior to the implementation of the first interventions, and/or the lack of a control district monitored simultaneously, the magnitude of the effect of vector control interventions on local vector population densities could not be quantified through robust statistical models. Future projects should include a baseline of at least one year, or a representative control district, to properly and quantitatively evaluate the effectiveness of the interventions on local vector populations. This is critical to identifying and subsequently addressing the gaps in the protection offered.

## Conclusions

*Anopheles arabiensis* was the main vector species during the Magude project. *Anopheles merus*, *An. parensis*, *An. funestus s.s.* and *An. squamosus* likely played a secondary and minor role. Further investigation into the possible role of other collected vector species (i.e. *Anopheles* species that are known secondary vectors elsewhere in Africa) is needed. The deployment of MDA and IRS, in addition to LLINs, successfully reduced vector sporozoite rates during the first year of implementation but no further reduction in sporozoite rates was observed despite similar intervention coverages in the second year. IRS most likely controlled *An. funestus s.s.* and *An. parensis* and was also effective at controlling *An. arabiensis*, but its effect was limited by its short residual efficacy that went below optimal levels (80% mosquito mortality in WHO cone bioassays) before the end of the high transmission season. Its effect on *An. merus* and *An. squamosus* could not be assessed due to low mosquito numbers but should be investigated as these species were incriminated as malaria vectors in Magude. LLINs complemented the protection provided by IRS, especially by providing protection against the indoor and late-evening host-seeking *An. arabiensis* and *An. merus* vectors. Therefore, the combination of IRS and LLIN is likely to have brought added value to the control of malaria vectors during the Magude project, compared to the implementation of one intervention alone. However, the effect of LLINs was compromised by their suboptimal use and the pyrethroid resistance in *An. funestus s.s.* and *An. parensis* populations. Future progress towards malaria elimination could be made by increasing LLIN use and distributing dual active ingredient LLINs to prevent transmission by the *An. arabiensis* and *An. merus* that survive IRS, by sustaining IRS to maintain control of *An. funestus s.s.* and *An. parensis*, and by using longer lasting residual insecticides for IRS to prevent the increase in vector densities observed at the end of the rainy season. Additional interventions will nevertheless be needed to tackle those vectors that transmit malaria outdoors and early indoors if we are to close the gap towards malaria elimination.

## Supporting information

**S1 File. Exclusions criteria for mosquito collections.**
(DOCX)

**S2 File. Sinusoidal function used to simulate ITN use based on observed values of ITN use.**
(DOCX)

**S3 File. Detailed Sporozoite detection results.**
(DOCX)

**S4 File. Detailed model results.**
(DOCX)

**S1 Data.**
(XLSX)

## Acknowledgments

We thank members of the Magude communities who allowed us to collect mosquitoes in their households. We also thank all the CISM entomology team members for their assistance during the study.

## Author Contributions

**Conceptualization:** Krijn P. Paaijmans.

**Data curation:** Lucía Fernández Montoya, Helena Martí-Soler.

**Formal analysis:** Lucía Fernández Montoya, Helena Martí-Soler, Ellie Sherrard-Smith.

**Investigation:** Lucía Fernández Montoya, Mara Máquina, Kiba Comiche, Inocencia Cuamba, Celso Alafo, Krijn P. Paaijmans.

**Methodology:** Lucía Fernández Montoya, Krijn P. Paaijmans.

**Project administration:** Lucía Fernández Montoya, Beatriz Galatas, Pedro Aide, Francisco Saúte, Krijn P. Paaijmans.

**Resources:** Nelson Cuamba, Dulcisaria Jotamo.

**Supervision:** Quique Bassat, Krijn P. Paaijmans.

**Validation:** Lizette L. Koekemoer, Ellie Sherrard-Smith.

**Visualization:** Lucía Fernández Montoya.

**Writing – original draft:** Lucía Fernández Montoya, Krijn P. Paaijmans.

**Writing – review & editing:** Helena Martí-Soler, Mara Máquina, Kiba Comiche, Inocencia Cuamba, Celso Alafo, Lizette L. Koekemoer, Ellie Sherrard-Smith, Quique Bassat, Beatriz Galatas, Pedro Aide, Nelson Cuamba, Dulcisaria Jotamo, Francisco Saúte, Krijn P. Paaijmans.

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
