## [Decision Letter · Decision Letter 0]

22 Jul 2022

PONE-D-22-18501The mosquito vectors that sustained malaria transmission during the Magude project despite the combined deployment of indoor residual spraying, insecticide-treated nets and mass-drug administration.PLOS ONE

Dear Dr. Fernandez Montoya,

Thank you for submitting your manuscript to PLOS ONE. After careful consideration, we feel that it has merit but does not fully meet PLOS ONE’s publication criteria as it currently stands. Therefore, we invite you to submit a revised version of the manuscript that addresses the relatively minor points raised during the review process - see below.

We look forward to receiving your revised manuscript.

Kind regards,

Basil Brooke, PhD

Academic Editor

PLOS ONE

Journal Requirements:

"We acknowledge support from the Spanish Ministry of Science and Innovation through the “Centro de Excelencia Severo Ochoa 2019-2023” Program (CEX2018-000806-S), and support from the Generalitat de Catalunya through the CERCA Program. CISM is supported by the Government of Mozambique and the Spanish Agency for International Development (AECID). LLK is supported by a DST/NRF South African Research Chairs Initiative Grant (UID 64763). ESS is funded by a UKRI Future Leaders Fellowship from the Medical Research Council (MR/T041986/1) and acknowledges funding from the MRC Centre for Global Infectious Disease Analysis (reference MR/R015600/1), jointly funded by the UK Medical Research Council (MRC) and the UK Foreign, Commonwealth & Development Office (FCDO), under the MRC/FCDO Concordat agreement and is also part of the EDCTP2 programme supported by the European Union; and acknowledges funding by Community Jameel."

"This study was supported by the Bill and Melinda Gates Foundation and Obra Social “la Caixa” Partnership for the Elimination of Malaria in Southern Mozambique (INV-0084830). LLK is supported by a DST/NRF South African Research Chairs Initiative Grant (UID 64763). ESS is funded by a UKRI Future Leaders Fellowship from the Medical Research Council (MR/T041986/1) and acknowledges funding from the MRC Centre for Global Infectious Disease Analysis (reference MR/R015600/1). Abt Associates Inc. provided support in the form of salaries for author NC, but did not have any additional role in the study design, data collection and analysis, decision to

publish, or preparation of the manuscript. This does not alter our adherence to PLOS ONE policies on sharing data and materials."

3. Please ensure that you refer to Figure 4 in your text as, if accepted, production will need this reference to link the reader to the figure.

4. Please upload a copy of Figure 6, to which you refer in your text on page 20. If the figure is no longer to be included as part of the submission please remove all reference to it within the text.

Reviewers' comments:

Reviewer's Responses to Questions

**Comments to the Author**

1. Is the manuscript technically sound, and do the data support the conclusions?

Reviewer #1: Yes

Reviewer #2: Yes

2. Has the statistical analysis been performed appropriately and rigorously? 

Reviewer #1: Yes

Reviewer #2: Yes

3. Have the authors made all data underlying the findings in their manuscript fully available?

Reviewer #1: Yes

Reviewer #2: Yes

4. Is the manuscript presented in an intelligible fashion and written in standard English?

Reviewer #1: Yes

Reviewer #2: Yes

5. Review Comments to the Author

Reviewer #1: This paper presents interesting results that are relevant for many malaria control programmes in southern/eastern Africa. My only major problem was that it seems very detailed and I wondered if it could be more focused, for example presenting the data in tables instead of in the text as is currently the case in a lot of the Results section.

The following are minor corrections.

Abstract

Line 57 – “b,y” should be “by,

Introduction

Line 102 – Ref #11 is a duplicate of ref #8. Delete #11.

Results

Line 386 – delete “with” at the very end of this line.

Lines 392-393 – why could these mosquitoes not be identified morphologically? Were they too damaged?

Line 395 – replace “were” with “to the”

Line 510 – the number of An. garnhami is missing. Also, check the spelling of “garnhami” throughout.

Line 511 – it should be An. listeri (no ‘a’ at the end). Also, as far as I know, neither An. multicolor or An. salbaii occur so far south. I would delete all mention of both of these species.

Line 513-514 – this is basically a repetition of what is said in the previous paragraph.

Lines 548-549 – An. quadriannulatus is mentioned twice and no mention is made of An. rufipes.

Line 553 – shouldn’t this be “Fig. 4”?

Discussion

Line 603 – reference #34 is about work in Benin, not South Africa.

References

Ref #

11 – This is a repeat of ref 8. Delete.

15-17 – There is no journal information. I assume that these three papers have been submitted to PLoS One but not yet published? If so, then include “(submitted)” at the end of each article.

18 – abbreviate the journal name as you have done for all the other refs.

21 – it should be “(Diptera: Culicidae)”.

22 – it should be “Brogdon”. Also, why are the first names given for the first two authors but not for Collins? You do not give first names for any other article.

39,40,46,55 – abbreviate the journal name.

41 – is this author’s name correct?

48, line 976 – give both ‘ anopheles’ a capital A.

Reviewer #2: Dear Authors, please consider the following suggested edits:

Page 12, line 133: You say: “…a rainy season expanding from October to March…” I suggest replace “expanding” with “extending”.

Page 14, Line 200: You say: “…The presence of sporozoites in individual mosquito samples was analyzed through screening enzyme-linked immunosorbent assays (ELISA) conducted on mosquitoes’ grinded head and thorax…”. It would be useful to know which parasites you assayed for, whether it was only for P falciparum or if you also tested for P vivax. Vivax is known to occur in the region and because the focus is usually on testing for falciparum, little appears to be known regarding vivax prevalence. So just clarify which parasites you tested for and why you excluded others.

Page 16, Line 266 – 275: You say “…The impact of both MDA and IRS on sporozoite rates can potentially already be observed two to three weeks after implementation. MDA with DHAp immediately eliminates gametocides from humans, which prevents feeding mosquitoes from ingesting gametocytes and becoming infective. The tim….”. This text is under “Data Analysis” and strictly speaking is probably more appropriate under Results or Discussion, but is probably not too serious. Just be more careful to appropriately allocate material more appropriately in future. I can see that you were intending to create context for your analysis but this could also be done under Results or Discussion.

Page 16, Line 270: You say “…migration to mosquito salivary glands can be of two or three weeks depen…”. I suggest delete the “of” before “two or three weeks”.

Page 19, Lines 353-356: You say “…We evaluated indoor and outdoor host-seeking times during the project’s intervention period (August 2015 to October 2017) by calculating the number of host-seeking mosquitoes of each species collected per person for each 2 hour time interval (period of rotation of the CDC bottle rotator) from 18:00 to 06:00, b…”. I just want to mention that earlier in the article you state that, indoors, the CDC traps were hung 1,5 metres above ground at the foot end of the beds, and therefore the primary attractant was human odour and thus the traps and numbers of mosquitoes caught only really become effective and valid once a person has gone to bed, presumably around 9pm. So one needs to exercise caution in apportioning host-seeking times based purely on trap catches when the “bait” is not in use in early evening.

Page 20, Lines 372-374: You say “…More details on the ethical considerations of the population census, household surveys, cross-sectional 374 prevalence surveys and MDAs are provided elsewhere…”. Where is “elsewhere? I don’t really care to be honest, and neither will most readers of the paper, but you cannot make a statement like this without giving some indication where the details are available; it is like saying some piece of information has been published “elsewhere” but you do not give the reference.

Page 20, Lines 394-395: You say “…and 87.3% 395 (n=332) of mosquitoes belonging were An. funestus group.”. Fix the language please, by deleting “were” and replacing with “to the”.

Page 24, Line 511: You say “…None of the An. listeria,…”, it should be An listeri.

Page 29, Line 642: Please italicize An. gambiae.

Page 31, Line 696: You say “…we continued to find large enough number of this species…” Either say “a large enough number” or “large enough numbers”.

Page 36. References 8 and 11 are repetitions of the same publication, you need to delete one.

Page 37. References 15, 16 and 17 have no year of publication or page numbers, please correct.

Page 40. Reference 48. The genus name needs to be capitalized.

Pge 45, Figure 2. “Indetermined” should be “Undetermined”. Same with Figures 3 and 5.

6. PLOS authors have the option to publish the peer review history of their article (what does this mean?). If published, this will include your full peer review and any attached files.

Reviewer #1: No

Reviewer #2: **Yes: **Leo Braack

---

## [Author Response · Author response to Decision Letter 0]

12 Aug 2022

Dear Editor,

We like to thank the two reviewers for their constructive comments and excellent attention to detail. We were very pleased to see that the reviewers had a very good understanding of the topic, which lead to the improvement of our manuscript. We have largely accepted the recommendations and have answered all questions below (in blue bold text).

We acknowledge that we cite three of our other papers that are currently in press (PONE-D-21-29269R2), in revision (PONE-D-21-40329), and in review (PONE-D-22-14587). We could work with PLOS ONE and e.g., publish those manuscripts on BioRXiv so we can properly cite them. But your suggestions are much appreciated.

Kind regards,

Lucia Fernandez Montoya, on behalf of all co-authors

Journal Requirements:

We feel our manuscript now adheres to PLOS ONE’s style requirements but do let us know if more changes are required. 

"We acknowledge support from the Spanish Ministry of Science and Innovation through the “Centro de Excelencia Severo Ochoa 2019-2023” Program (CEX2018-000806-S), and support from the Generalitat de Catalunya through the CERCA Program. CISM is supported by the Government of Mozambique and the Spanish Agency for International Development (AECID). LLK is supported by a DST/NRF South African Research Chairs Initiative Grant (UID 64763). ESS is funded by a UKRI Future Leaders Fellowship from the Medical Research Council (MR/T041986/1) and acknowledges funding from the MRC Centre for Global Infectious Disease Analysis (reference MR/R015600/1), jointly funded by the UK Medical Research Council (MRC) and the UK Foreign, Commonwealth & Development Office (FCDO), under the MRC/FCDO Concordat agreement and is also part of the EDCTP2 programme supported by the European Union; and acknowledges funding by Community Jameel."

"This study was supported by the Bill and Melinda Gates Foundation and Obra Social “la Caixa” Partnership for the Elimination of Malaria in Southern Mozambique (INV-0084830). LLK is supported by a DST/NRF South African Research Chairs Initiative Grant (UID 64763). ESS is funded by a UKRI Future Leaders Fellowship from the Medical Research Council (MR/T041986/1) and acknowledges funding from the MRC Centre for Global Infectious Disease Analysis (reference MR/R015600/1). Abt Associates Inc. provided support in the form of salaries for author NC, but did not have any additional role in the study design, data collection and analysis, decision to publish, or preparation of the manuscript. This does not alter our adherence to PLOS ONE policies on sharing data and materials."

We have edited the Acknowledgment statement to remove any reference to funding sources. 

Please update our funding statement as follows:

"This study was supported by the Bill and Melinda Gates Foundation and Obra Social “la Caixa” Partnership for the Elimination of Malaria in Southern Mozambique (INV-008483). LLK is supported by a DST/NRF South African Research Chairs Initiative Grant (UID 64763). ESS is funded by a UKRI Future Leaders Fellowship from the Medical Research Council (MR/T041986/1) and acknowledges funding from the MRC Centre for Global Infectious Disease Analysis (reference MR/R015600/1), jointly funded by the UK Medical Research Council (MRC) and the UK Foreign, Commonwealth & Development Office (FCDO), under the MRC/FCDO Concordat agreement and is also part of the EDCTP2 programme supported by the European Union; and acknowledges funding by Community Jameel. Abt. Associates Inc. provided support in the form of salaries for author NC, but did not have any additional role in the study design, data collection and analysis, decision to publish, or preparation of the manuscript. This does not alter our adherence to PLOS ONE policies on sharing data and materials. The specific role of this author is articulated in the ‘author contributions’ section."

3. Please ensure that you refer to Figure 4 in your text as, if accepted, production will need this reference to link the reader to the figure.

We have inserted a reference to this figure at the end of the results section “Association between host-seeking Anopheles per person and residual malaria incidence”, line 554.

4. Please upload a copy of Figure 6, to which you refer in your text on page 20. If the figure is no longer to be included as part of the submission please remove all reference to it within the text.

There is actually no Figure 6, and we mistakenly wrote Fig 6 instead of Fig 4 at the end of the results section “Association between host-seeking Anopheles per person and residual malaria incidence”. We have corrected this.

We created the map ourselves using administrative boundaries from the Humanitarian Data Exchange under license "Creative commons attribution for Intergovernmental organizations. (CC-BY-IGO). https://data.humdata.org/faqs/licenses.

We have indicated this in the caption of Fig 1. 

We have revised all references to ensure that they meet PLOS ONE criteria. In addition, we have removed two references (former references number 11 and 34) and added one (number 23) to respond to reviewer’s comments. We cite three of our other papers that are currently in press (PONE-D-21-29269R2), in revision (PONE-D-21-40329), and in review (PONE-D-22-14587). We would like to work with PLOS ONE to find a solution so that we can cite them properly (e.g publish those manuscripts on BioRXiv). Your suggestions will be much appreciated.

***

Reviewers' comments:

Reviewer #1: This paper presents interesting results that are relevant for many malaria control programmes in southern/eastern Africa. My only major problem was that it seems very detailed and I wondered if it could be more focused, for example presenting the data in tables instead of in the text as is currently the case in a lot of the Results section. Thanks a lot for this suggestion. At the time of writing the manuscript, we tried to present the results of the first four paragraph of section “Anopheline species composition and densities over time” in different tables and graphs. Most of the co-authors found Fig 2 easier to interpret and more visually appealing than any other tables. Hence, we prefer to keep the results as they are now.

The following are minor corrections.

Abstract

Line 57 – “b,y” should be “by”, This is corrected.

Introduction

Line 102 – Ref #11 is a duplicate of ref #8. Delete #11. We have deleted the duplicate reference. 

Results

Line 386 – delete “with” at the very end of this line. Done.

Lines 392-393 – why could these mosquitoes not be identified morphologically? Were they too damaged? This is because they were either too damaged or because the microscopists could not find a matching species in the used taxonomy keys. We have added this sentence in line 393 of the results section.

Line 395 – replace “were” with “to the” Corrected.

Line 510 – the number of An. garnhami is missing. Also, check the spelling of “garnhami” throughout. The number of An. garnhami tested for P. falciparum sporozoites is 1. We have added it to line 510. We have also checked the spelling of “garnhami” everywhere and corrected as needed. 

Line 511 – it should be An. listeri (no ‘a’ at the end). Also, as far as I know, neither An. multicolor or An. salbaii occur so far south. I would delete all mention of both of these species. We have corrected An. listeri throughout the text. We prefer to keep mention to An. multicolor and An. salbaii in the manuscript, as An. multicolor and An. salbaii have been detected in equatorial and eastern Africa and we cannot rule out the possibility they inhabit southern Mozambique. In the discussion we are nuancing this by mentioning that the detection of these other Anopheles species could be an artifact of wrong morphological identification and that molecular validation will be required.

Line 513-514 – this is basically a repetition of what is said in the previous paragraph. The previous paragraph provides the total numbers and percentages of Pf sporozoite positive mosquitoes while this particular paragraph provides the percentage for indoor collected mosquitoes only. We reckon that since most of the mosquitos were collected indoors, the percentages are very similar, but we want to provide the figures for both indoors and outdoors separately, so we have kept both paragraphs. 

Lines 548-549 – An. quadriannulatus is mentioned twice and no mention is made of An. rufipes. Thanks for spotting this. An. rufipes is associated at 10% significance. We have corrected this.

Line 553 – shouldn’t this be “Fig. 4”? Indeed, this should be Fig. 4, we have corrected it.

Discussion

Line 603 – reference #34 is about work in Benin, not South Africa. Thanks for spotting this mistake. We have removed reference #34

References

Ref #

11 – This is a repeat of ref 8. Delete. We have deleted reference 11 and corrected reference numbers along the manuscript accordingly. 

15-17 – There is no journal information. I assume that these three papers have been submitted to PLoS One but not yet published? If so, then include “(submitted)” at the end of each article. We cite three of our other papers that are currently in press, in revision and in review respectively. We will work with PLOS ONE to find the best solution to cite these papers properly (e.g., publishing those manuscripts on BioRXiv). 

18 – abbreviate the journal name as you have done for all the other refs. Corrected.

21 – it should be “(Diptera: Culicidae)”. Corrected.

22 – it should be “Brogdon”. Also, why are the first names given for the first two authors but not for Collins? You do not give first names for any other article. This was a mistake. We have corrected it to provide only surname and name’s initials. 

39,40,46,55 – abbreviate the journal name. We have revised all reference to make sure that the journal names are abbreviated. 

41 – is this author’s name correct? Corrected.

48, line 976 – give both ‘ anopheles’ a capital A. Corrected.

Reviewer #2: Dear Authors, please consider the following suggested edits:

Page 12, line 133: You say: “…a rainy season expanding from October to March…” I suggest replace “expanding” with “extending”. Thanks a lot for this suggestion. We have replaced “expanding” for “extending”.

Page 14, Line 200: You say: “…The presence of sporozoites in individual mosquito samples was analyzed through screening enzyme-linked immunosorbent assays (ELISA) conducted on mosquitoes’ grinded head and thorax…”. It would be useful to know which parasites you assayed for, whether it was only for P falciparum or if you also tested for P vivax. Vivax is known to occur in the region and because the focus is usually on testing for falciparum, little appears to be known regarding vivax prevalence. So just clarify which parasites you tested for and why you excluded others. We tested for P. falciparum only because we were expecting very few sporozoite positive mosquitoes (given the elimination context) and P. falciparum accounts for 90% of all infections in Mozambique and almost all in the district neighboring the study area. We have edited the sentence to add the parasite species that we looked for, i.e. “… The presence of P. falciparum sporozoites in individual mosquito samples was analyzed through screening enzyme-linked immunosorbent assays (ELISA) conducted on mosquitoes’ grinded head and thorax ...” and the justification for testing only this species, i.e. “The presence of sporozoites of other Plasmodium species was not tested because P. falciparum is known to account for over 90% of all diagnosed malaria infections in Mozambique [23] and for almost all in the neighboring district of Manhiça [24], and because very low positivity rates were expected given the elimination context.”

Page 16, Line 266 – 275: You say “…The impact of both MDA and IRS on sporozoite rates can potentially already be observed two to three weeks after implementation. MDA with DHAp immediately eliminates gametocides from humans, which prevents feeding mosquitoes from ingesting gametocytes and becoming infective. The tim….”. This text is under “Data Analysis” and strictly speaking is probably more appropriate under Results or Discussion, but is probably not too serious. Just be more careful to appropriately allocate material more appropriately in future. I can see that you were intending to create context for your analysis but this could also be done under Results or Discussion.

Thanks for this suggestion. Indeed, we want to provide rationale for the time lags used in the analysis. We tried to move it to the result section, but some co-authors have found it confusing to scroll up and down through the manuscript to get this information. Consequently, we have decided to leave it in the data analysis section.

Page 16, Line 270: You say “…migration to mosquito salivary glands can be of two or three weeks depen…”. I suggest delete the “of” before “two or three weeks”. Thanks for the suggestion. We have deleted “of”. 

Page 19, Lines 353-356: You say “…We evaluated indoor and outdoor host-seeking times during the project’s intervention period (August 2015 to October 2017) by calculating the number of host-seeking mosquitoes of each species collected per person for each 2 hour time interval (period of rotation of the CDC bottle rotator) from 18:00 to 06:00, b…”. I just want to mention that earlier in the article you state that, indoors, the CDC traps were hung 1,5 metres above ground at the foot end of the beds, and therefore the primary attractant was human odour and thus the traps and numbers of mosquitoes caught only really become effective and valid once a person has gone to bed, presumably around 9pm. So one needs to exercise caution in apportioning host-seeking times based purely on trap catches when the “bait” is not in use in early evening. Thanks so much for raising this point. It is a very important observation. We have added it to the limitations section of the discussion.

Page 20, Lines 372-374: You say “…More details on the ethical considerations of the population census, household surveys, cross-sectional 374 prevalence surveys and MDAs are provided elsewhere…”. Where is “elsewhere? I don’t really care to be honest, and neither will most readers of the paper, but you cannot make a statement like this without giving some indication where the details are available; it is like saying some piece of information has been published “elsewhere” but you do not give the reference. Thanks for spotting this missing reference. We have added reference 12 and 13 here, which provide all details on the census, household surveys, cross-sectional prevalence surveys and MDAs.

Page 20, Lines 394-395: You say “…and 87.3% 395 (n=332) of mosquitoes belonging were An. funestus group.”. Fix the language please, by deleting “were” and replacing with “to the”. Corrected.

Page 24, Line 511: You say “…None of the An. listeria,…”, it should be An listeri. Corrected.

Page 29, Line 642: Please italicize An. gambiae. Corrected.

Page 31, Line 696: You say “…we continued to find large enough number of this species…” Either say “a large enough number” or “large enough numbers”. Corrected.

Page 36. References 8 and 11 are repetitions of the same publication, you need to delete one. Corrected.

Page 37. References 15, 16 and 17 have no year of publication or page numbers, please correct.

We cite three of our other papers that are currently in press, in revision and in review respectively. We will work with PLOS ONE to find the best solution to cite these papers properly e.g., publishing those manuscripts on BioRXiv so we can properly cite them. 

Page 40. Reference 48. The genus name needs to be capitalized. Corrected.

Pge 45, Figure 2. “Indetermined” should be “Undetermined”. Same with Figures 3 and 5.We have corrected all three figures.

---

## [Editor Report · Decision Letter 1]

24 Aug 2022

The mosquito vectors that sustained malaria transmission during the Magude project despite the combined deployment of indoor residual spraying, insecticide-treated nets and mass-drug administration.

PONE-D-22-18501R1

Dear Dr. Fernandez Montoya,

We’re pleased to inform you that your manuscript has been judged scientifically suitable for publication and will be formally accepted for publication once it meets all outstanding technical requirements.

Kind regards,

Basil Brooke, PhD

Academic Editor

PLOS ONE
---

## [Editor Report · Acceptance letter]

31 Aug 2022

PONE-D-22-18501R1 

The mosquito vectors that sustained malaria transmission during the Magude project despite the combined deployment of indoor residual spraying, insecticide-treated nets and mass-drug administration. 

Dear Dr. Fernandez Montoya:

I'm pleased to inform you that your manuscript has been deemed suitable for publication in PLOS ONE. Congratulations! Your manuscript is now with our production department. 

Kind regards, 

on behalf of

Dr Basil Brooke 

Academic Editor

PLOS ONE